# Durvalumab plus tremelimumab for the treatment of advanced neuroendocrine neoplasms of gastroenteropancreatic and lung origin

Single immune checkpoint blockade in advanced neuroendocrine neoplasms (NENs) shows limited efficacy; dual checkpoint blockade may improve treatment activity. Dune (NCT03095274) is a non-randomized controlled multi-cohort phase II clinical trial evaluating durvalumab plus tremelimumab activity and safety in advanced NENs. This study included 123 patients presenting between 2017 and 2019 with typical/atypical lung carcinoids (Cohort 1), G1/2 gastrointestinal (Cohort 2), G1/2 pancreatic (Cohort 3) and G3 gastroenteropancreatic (GEP) (Cohort 4) NENs; who progressed to standard therapies. Patients received 1500 mg durvalumab and 75 mg tremelimumab for up to 13 and 4 cycles (every 4 weeks), respectively. The primary objective was the 9-month clinical benefit rate (CBR) for cohorts 1-3 and 9-month overall survival (OS) rate for Cohort 4. Secondary endpoints included objective response rate, duration of response, progression-free survival according to irRECIST, overall survival, and safety. Correlation of PD-L1 expression with efficacy was exploratory. The 9-month CBR was 25.9%/35.5%/25% for Cohorts 1, 2, and 3 respectively. The 9-month OS rate for Cohort 4 was 36.1%, surpassing the futility threshold. Benefit in Cohort 4 was observed regardless of differentiation and Ki67 levels. PD-L1 combined scores did not correlate with treatment activity. Safety profile was consistent with that of prior studies. In conclusion, durvalumab plus tremelimumab is safe in NENs and shows modest survival benefit in G3 GEP-NENs; with one-third of these patients experiencing a prolonged OS.

Neuroendocrine neoplasms (NENs) constitute a heterogeneous group of rare malignancies that arise from the diffuse neuroendocrine cell system and most frequently occur in the gastroenteropancreatic (GEP) tract and lung[1–4].

Systemic treatment for advanced well-differentiated grade 1 and 2 NENs includes somatostatin analogues, interferon, radionuclides, chemotherapy, targeted kinase inhibitors (TKIs) such as sunitinib, and everolimus[4–8]. Stable disease (SD) is the most common treatment outcome, with patients progressing after variable time lapses. Vascular endothelial growth factor (VEGF) inhibitors have been investigated as potential therapies, considering the remarkable vascular dependence of NENs[9,11]. For high-grade NENs, first-line treatment with cisplatin or carboplatin and etoposide combinations is well established, and has reported positive antitumor activity. However, the impact on survival is limited and patient life expectancy remains under 12 months[12–14].

✉ e-mail: jcapdevila@vhio.net

Immune checkpoint inhibition (ICI) of programmed death ligand 1 (PD-L1) and cytotoxic T-lymphocyte antigen 4 (CTLA-4) has changed the paradigm in many cancer types[15,16]. Nevertheless, the role of immunotherapy in NENs remains controversial. Single agents targeting PD-L1 reported limited antitumor activity in NENs, particularly for well-differentiated neuroendocrine tumours (NETs)[17–21]. Only patients with lung-NETs have achieved a higher objective response rate (ORR) of 20%, regardless of PD-L1 expression[20]. Conversely, toripalimab reported promising activity that was favourably associated with PD-L1–positive expression and high tumour mutational burden (TMB) regardless of tumour origin[21].

Currently, the dual targeting of PD-L1 and CTLA-4 seems to overcome resistance to single-agent immunotherapy in other cancer types[15,16]. Early reports from the combination of nivolumab plus ipilimumab showed controversial antitumor activity in NECs[22–24].

In this work, we aim to evaluate the potential activity and safety of the combination of durvalumab (anti-PD-L1) and tremelimumab (anti-CTLA-4) in specific cohorts of patients with advanced NENs (DUNE).

## Results

### Baseline patient characteristics

Between April 2017 and December 2019, 123 patients were enroled in the study (Fig. 1A). NEN distribution was as follows: 27 typical or atypical lung carcinoids (Cohort 1); 31 G1-2 gastrointestinal (Cohort 2); 32 G1-2 pancreatic (Cohort 3); and 33 high-grade (grade 3) GEP (Cohort 4). All patients received at least one dose of durvalumab plus tremelimumab (Fig. 1A and Supplementary Table 1).

### Activity endpoints

Overall CBR according to RECIST 1.1 was 56.1% (95% CI: 47.3–64.6), and 66.7% (95% CI: 47.9–82.1), 74.2% (95% CI: 57.1–87.0), 59.4% (95% CI: 42.2–75.0), and 27.3% (95% CI: 14.4–43.9) for all included patients and Cohorts 1 to 4, respectively (Fig. 1B–E). The 9-m CBR was 25.9% (95% CI: 12.4–44.3), 35.5% (95% CI: 20.5–53.0), 25% (95% CI: 12.6–41.7) and 6.1% (95% CI: 1.3–18.1) for Cohorts 1 to 4, respectively (Supplementary Table 2). In total, 1 (0.8%) patient had CR, 7 (5.7%) PR and 61 (49.6%) SD as their best response (Supplementary Table 2 and Fig. 1B–E). In lung-NETs, PR was reported in 2 (33.3%) patients with atypical carcinoids and 1 (4.7%) with typical carcinoids. In patients with grade 3 NENs, NECs had an ORR of 16.7% (3/18 patients) versus 0% in NETs. No significant differences were found between evaluations according to RECIST 1.1 or irRECIST 1.1 within any cohort (Supplementary Table 2). The median DoR was 10.4 months (95% CI: 2.7–24.3), while SD was maintained during a median time of 5.5 months (95% CI: 3.4–6.1) (Supplementary Fig. 1). There was no statistically significant correlation between response and PD-L1 expression by CPS (Supplementary Table 3). Microsatellite instability was assessed in 7 out of 8 patients who had a response, being all microsatellite stable.

With a median follow-up of 16.5 months (range: 0.3-42.9), the median PFS was 5.6 (95% CI: 4.9–6.2), 5.8 (95% CI: 3.1–8.5), 5.5 (95% CI: 2.4–8.7) and 2.4 (95% CI: 2.1–2.8) months, in cohorts 1 to 4 respectively (Fig. 2A). No statistically significant differences were found in PFS based on PD-L1 CPS (Fig. 2B–E).

A total of 77 (62.6%) patients died throughout the study period, due to disease progression 69 (56.1%), toxicity 3 (2.4%), clinical deterioration 1 (0.8%), carcinoid crisis 1 (0.8%), progression of secondary neoplasm 1 (0.8%), cerebrovascular incident 1 (0.8%), and non-coronavirus disease pneumonia 1 (0.8%). The median OS was not reached (range: 0.3–41.3) in lung-NETs (Cohort 1) and was 29.5 (95% CI: 19.6–39.4), 23.8 (95% CI: 16.4–31.2), and 5.9 (95% CI: 2–9.7) months for cohorts 2 to 4, respectively (Fig. 3). For high-grade GEP-NENs (Cohort 4), the 9-m OS rate, which was the primary endpoint, was 36.1% (95% CI: 19.6–52.6), surpassing the pre-established futility threshold (23%; H0 = 13%). Moreover, 10 (30.3%) patients with GEP-NENs surpassed the 12 months survival (long-survivors). A stratified

analysis for OS within grade 3 GEP-NENs found no significant associations between survival status and baseline characteristics, including histological differentiation (NET vs NEC) or PD-L1 CPS status (Supplementary Table 4).

### Safety

Overall, only 16 (13%) patients completed the treatment as initially scheduled and treatment was discontinued prematurely in most patients (87%) mainly due to: the progression of the disease 74 (60.2%), unacceptable toxicity 12 (9.8%), death 6 (4.9%) or physician criteria 4 (3.3%) (Fig. 1A). The median number of administered cycles was 5 for durvalumab and 4 for tremelimumab.

Most treatment-related adverse events (TRAEs) were mild and resolved with appropriate clinical care; the most common TRAEs across the cohorts were: fatigue (44.7%), diarrhoea (32.5%) and pruritus (23.6%) (Supplementary Table 5). Grade ≥3 toxicities had low frequency (29.3% of patients), and the most common were diarrhoea (6.5%), transaminitis (4.9%), fatigue (3.3%), and vomiting (2.4%) (Supplementary Table 5). Grade ≥3 immune-related AEs occurred in 12.2% of patients and included as the more common events hepatitis (1.6%), myositis (1.6%) and anaemia (1.6%). Deaths caused by toxicity were associated with hepatic failure, myasthenia gravis and diarrhoea that worsened and coursed with a potential encephalitis infection.

## Discussion

Treatment with durvalumab plus tremelimumab showed modest antitumor activity in this large cohort ($n = 123$) of heavily pre-treated patients with NEN, regardless of origin, histological grade, differentiation or PD-L1 expression.

The long-term clinical benefit rate was chosen as the primary endpoint to include long-term stabilisation of the disease as a therapeutic success. The overall CBR (56.1% by RECIST 1.1) was not significantly better than that previously reported with a single ICI with pembrolizumab (59.8%) or spartalizumab (63.2%)[17,20]. Response to treatment was documented only in eight (6.5%) patients, and was comparable to that reported for single ICI, or dual ICI with an ORR of 14.9% in advanced lung or GEP-NENs[17–20,23]. Patients with lung-NENs were previously identified as a potentially promising group for immunotherapy, with ORR ranging from 18.2 to 20%, up to 33% in atypical carcinoids[19,22,24]. Conversely, patients with lung-NETs in our study had an ORR of 11.1%, indicating limited antitumoral activity for durvalumab plus tremelimumab. Despite the low number of responses, we did observe a higher response rate in patients with atypical carcinoids in agreement with previous trials[22,24]. Our results showed no enrichment of activity regarding histological grade, with modest activity also in high-grade NENs (ORR 9.1%), with all responses occurring in patients with NECs. This was concordant with early studies of nivolumab plus ipilimumab, showing an ORR of up to 44% in patients with high-grade NECs previously treated with chemotherapy[22]. Due to the indirect nature of the comparison, our findings should be interpreted with caution when compared to those of previous trials, which may include low- or high-risk patients. For example, the DART trial excluded pancreatic NENs[22]. Moreover, durvalumab targets PD-L1 while nivolumab is an anti-PD-1 antibody and given additional receptor-ligand interactions, immunological and clinical outcomes may differ between these treatment regimens.

DoR achieved by immunotherapy in our trial and in previous reports involving low-grade NENs was shorter than that recently reported with targeted therapies such as lenvatinib, with a median DoR of 21.5 months (range: 8.4–38.3)[11].

The median overall PFS was 5.3 months, which is in the range of that reported with single-agent pembrolizumab after progression to standard therapies (median 4.1 months; 95% CI: 3.5–5.4)[17]; or nivolumab plus ipilimumab (median 4 months; 95% CI: 3–6)[23].

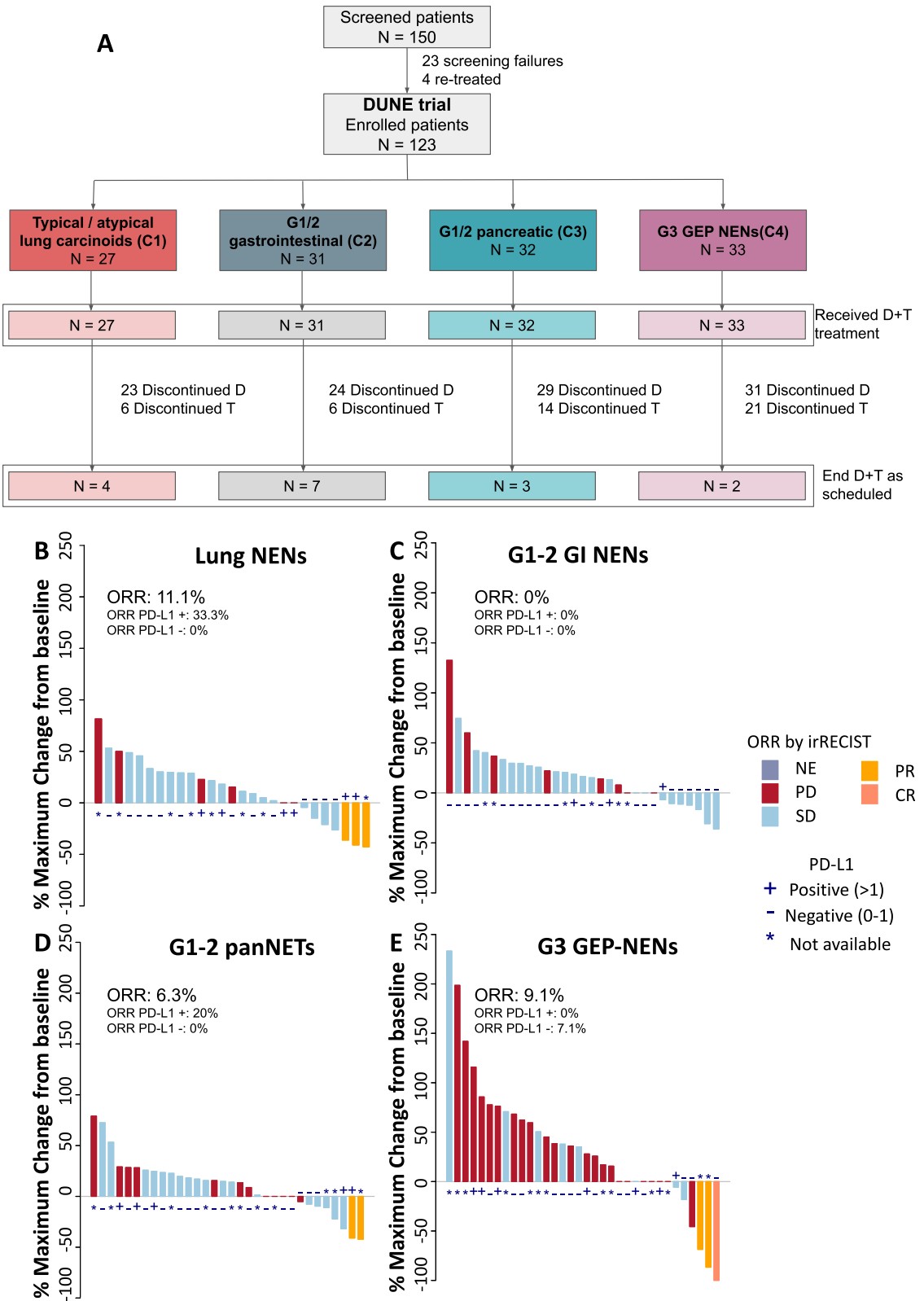

**Fig. 1 | Patient flowchart and ORR for each cohort.** Patient flowchart (**A**) and ORR for each cohort. **B** Typical/atypical lung carcinoids; **C** Low-grade (Grade 1–2) gastrointestinal NENs; **D** Low-grade (Grade 1–2) pancreatic NENs; **E** High-grade (grade 3) gastroenteropancreatic NENs. ORR in PD-L1 CPS subgroups were calculated regarding those patients with evaluable CPS scores. Source data are provided as a Source Data file. NENs neuroendocrine neoplasms, GEP Gastroenteropancreatic, CR complete response, GI-NET GI neuroendocrine tumour, ORR overall response rate, PD progressive disease, PR partial response, SD stable disease, PD-L1 Programmed death ligand 1.

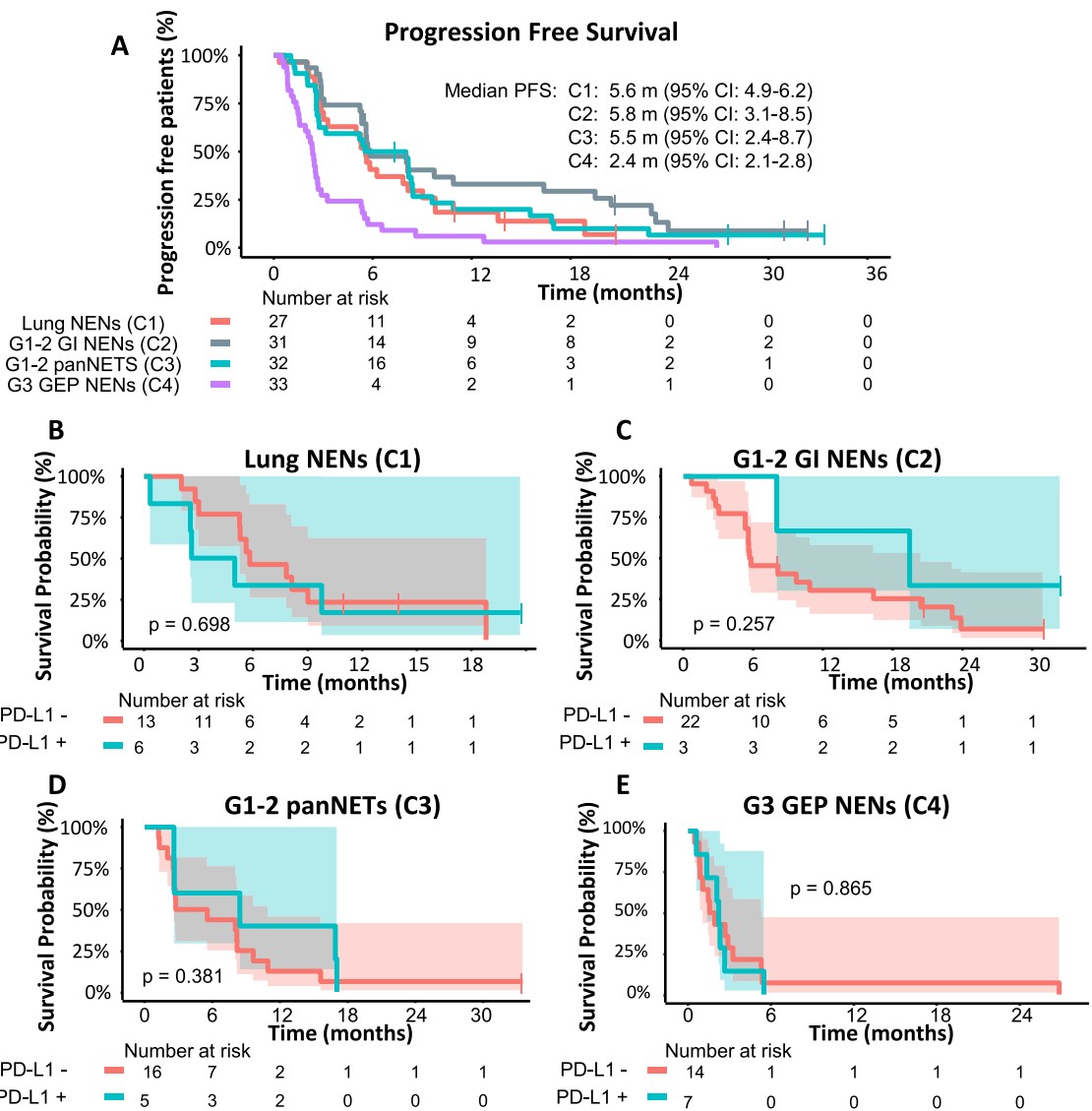

**Fig. 2 | PFS for each cohort and PFS stratified by PD-L1 CPS status.** PFS for each cohort (**A**) and PFS stratified by PD-L1 CPS status for each cohort (**B**–**E**). Shadowed areas represent the confidence intervals for PFS. Statistical comparison by log-rang.

NENs neuroendocrine neoplasms, PFS progression-free survival. Source data are provided as a Source Data file.

Two trials in patients with low-grade NENs with single-agent pembrolizumab reported a median OS between 21 and 24 months, which was in the range of the OS among patients with low-grade NENs in our study, suggesting a small benefit with the addition of CTLA-4 blockade[17,18]. Previous trials using dual ICI also failed to improve survival[22,23]. In high-grade NENs, durvalumab plus treme-limumab showed a modest improvement in survival, surpassing the pre-established futility threshold. Ten patients achieved prolonged survival, longer than 12 months, suggesting the potential use of dual ICI in a selected subtype of patients within this setting. Long-term survivors had mostly poorly differentiated NECs (70%) by central review; however, no significant correlation was found between baseline molecular or clinical biomarkers and treatment in this subgroup.

The differences observed in the efficacy and activity of immu-notherapy between low- and high-grade NENs across trials may rely on the higher PD-L1 expression, TMB, and enhanced neoantigen pre-sentation, which has been positively correlated to tumour grade[25]. Tumour PD-L1 expression has been associated with ICI efficacy across tumour types and positively associated with poorer survival in NENs[26]. However, we did not observe a correlation between PD-L1 CPS with

efficacy. A further explanation could be the potential immunogenic effect attributable to chemotherapy, which is the standard first-line treatment for high-grade GEP-NENs. Platinum-based chemotherapy is capable of modulating tumour-infiltrating lymphocytes (TILs) and reactivating antitumor immunity within an immunosuppressive microenvironment[27,28]. In fact, two phase III trials demonstrated ben-efit in survival with the addition of durvalumab or atezolizumab to first-line platinum-etoposide[29,30]. Based on these findings, the admin-istration of dual ICI in combination with standard first-line che-motherapy may be considered a reasonable option to explore in high-grade NENs. Currently, the phase 2 trial NICENEC explores the role of nivolumab in combination with platinum-based chemotherapy as first line for the treatment of patients with high-grade GEP-NENs and reported promising activity and, similarly to our results, prolonged survival in a subset of patients[31].

Regarding safety, our findings are consistent with those of pre-vious reports[32,33]. Premature treatment discontinuation due to toxicity was required in about 10% of patients, which unlikely impacted activ-ity. The relatively short treatment exposure may explain the low inci-dence of severe immune-related adverse events compared with other trials with combined checkpoint blockade.

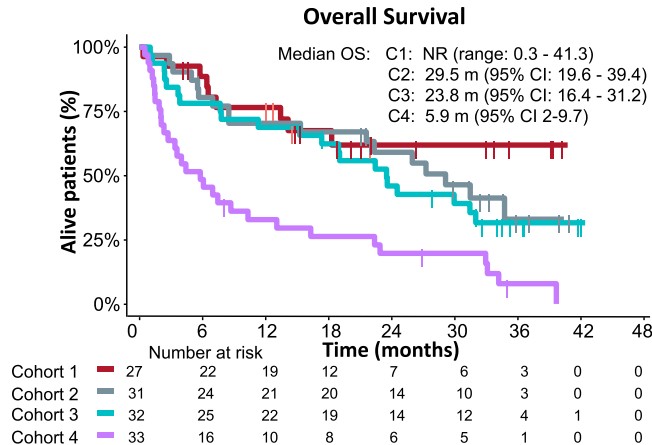

**Fig. 3 | OS for each cohort.** Cohort 1, typical or atypical lung carcinoids (red); Cohort 2, G1-2 gastrointestinal (grey); Cohort 3 G1-2 pancreatic (green); and Cohort 4 high-grade (grade 3) GEP-NENs (purple). Source data are provided as a Source Data file. NENs neuroendocrine neoplasms, OS overall survival.

The main limitation of the DUNE trial was the lack of randomisation and a parallel control group comparing single ICI or alternative treatment options. There is limited information on prognosis and survival in such a heavily pre-treated population of NENs, which may have led to an overestimation of the expected primary endpoints. Despite the overall sample size being relevant, the sample size for each cohort limited the exploratory research of potential prognostic factors in the stratified analysis. The small sample size also limited the comparison between G3 NET vs NEC histology, which are tumours with very different behaviour. However, a centralised review by an experienced pathologist showed that long-term survivors included both NECs and NETs. The lack of a centralised review for the tumour assessment was another caveat. In contrast, histological differentiation and PD-L1 CPS were centrally reviewed, and differences in ORR attributable to the use of RECIST 1.1 or irRECIST criteria were excluded.

In summary, our findings support a potential modest use of dual ICI in high-grade NENs and atypical lung carcinoids. Further research with ICI in this setting may focus on long-term survival endpoints, and potentially shifting to a first-line setting in combination with chemotherapy. Prognostic and predictive biomarkers need to be further characterised in patients treated with ICI.

## Methods

### Study design and patients
This study was conducted in accordance with the principles of the Declaration of Helsinki and the International Conference on Harmonisation Guidelines for Good Clinical Practice. The study protocol (see Supplementary Note in the Supplementary Information file) was approved in the first instance in 2017 by the competent authority in Spain and the Independent Ethics Committee from Vall d'Hebron University Hospital. Written informed consent was obtained from all patients.

DUNE (EudraCT: 2016-002858-20; NCT03095274) is a prospective, single-arm, open-label, multicohort, multicentre, phase II trial involving 20 institutions in Spain. Eligible patients presented with a histologically confirmed diagnosis of advanced/metastatic NEN and had progressed to standard anticancer therapies according to tumour type. Patients were enrolled in four cohorts according to the type of NET: well-moderately differentiated NETs of the lung, also known as typical and atypical lung carcinoids, that have progressed to prior somatostatin analogues therapy, one prior targeted therapy or chemotherapy (Cohort 1); well-moderately differentiated, World Health

Organisation (WHO) grade 1 and 2, gastrointestinal NETs after progression to somatostatin analogues and one targeted therapy, interferon or radionuclides (Cohort 2); well-moderately differentiated, WHO grade 1 and 2 NET, from pancreatic origin after progression to at least two and a maximum of four standard therapies, including chemotherapy, somatostatin analogues and target therapy (Cohort 3); WHO grade 3 NENs, of gastroenteropancreatic or unknown primary origin, excluding lung primary carcinomas, after progression to first-line chemotherapy with a platinum-based regimen (Cohort 4). The trial used the WHO 2010 classification for NETs. General inclusion criteria also included patients >18 years; Eastern Cooperative Oncology Group performance status (ECOG PS) 0–1; life expectancy >12 weeks; adequate haematologic, hepatic, and renal function; measurable disease according to Response Criteria in Solid Tumours (RECIST) version 1.1[24]; and documented radiological disease progression according to RECIST 1.1 within 12 months prior to inclusion. The exclusion criteria were as follows: prior treatment with anti-PDL-1/anti-programmed death 1 (PD-1) or anti-CTLA-4 therapy; immunodeficiency or use of immunosuppressive medication history within 28 days before the first dose of durvalumab or tremelimumab, with the exception of intranasal and inhaled corticosteroids or systemic corticosteroids at physiological doses not exceeding 10 mg/day of prednisone, or equivalent; active or prior documented autoimmune disease within the past 2 years; previous or active interstitial lung disease, or non-infectious pneumonitis; presence of active brain metastases or secondary malignancies.

Accrual started on and finished on April 12th, 2017 and finished on November 30th, 2019. Accrual was competitive and sequential. The trial was registered on www.clinicaltrials.gov on March 29th, 2017, prior to patient inclusions.

### Procedures
Patients received intravenous durvalumab at a fixed dose of 1500 mg over a 4-week (Q4W) cycle for up to 13 cycles and intravenous tremelimumab at a fixed dose of 75 mg Q4W for up to four cycles. Treatment was administered until the completion of the treatment schedule (i.e. 12 months), confirmed progression of the disease, unacceptable toxicity, patient withdrawal or death; whichever occurred first. Dose reductions were not permitted, although doses could be delayed for up to 12 weeks due to toxicity. Patients could be retreated with durvalumab and tremelimumab in case of progression after completing the 4 cycles of durvalumab plus tremelimumab combination or at the end of the study treatment as scheduled while maintaining clinical benefit, according to the investigator's criteria.

Clinical assessments included medical history review; complete physical examination, including Eastern Cooperative Oncology Group Performance Status (ECOG PS), vital signs, height and weight; laboratory tests and urinalysis; record of adverse events (AEs); treatment compliance; and tumour marker assessments. Tumour imaging assessments by computed tomography (CT) or magnetic resonance imaging (MRI) scan at baseline, and every 12 weeks (±1 week) until disease progression or the initiation of an alternative treatment. CT or MRI scans were assessed locally by investigators following both, RECIST 1.1 and irRECIST 1.1 criteria. Histology (only cohort 4) and PD-L1 expression in archival formalin-fixed tumour samples was independently assessed by a central laboratory. PD-L1 combined positive score (CPS) was assessed using the PD-L1 IHC 22C3 pharmDx assay (Agilent Technologies, Carpinteria, CA, USA). PD-L1 antibody was from Roche (Cat n°: 741-4905). PD-L1 CPS of 1 or greater was considered positive. Central revision of microsatellite instability (MSI) phenotype by IHC was performed for patients who had a response. MSI phenotype diagnosis followed ESMO recommendations: loss of nuclear expression of at least one out of four proteins in IHC test performed using regular MLH1 (Roche, cat n°: 760-5091), MSH6 (Roche, cat n°: 760-

5092), PMS2 (Roche, cat n°: 760-5094), MSH2 (Roche, cat n°: 760-5093) antibodies.

## Outcome measures

The primary endpoint for Cohorts 1–3 was the 9-month (9-m) clinical benefit rate (CBR) assessed by local investigators according to RECIST 1.1, defined as the factual percentage of patients achieving CR, partial response (PR), or SD at 9-m after the initiation of durvalumab plus tremelimumab treatment. The primary endpoint for Cohort 4 was the 9-m overall survival (OS) rate, defined as the percentage of patients alive at 9-m after initiation of durvalumab plus tremelimumab therapy. The primary endpoint for each cohort was chosen based on the stipulated benchmarks[4–14] that reflected therapeutic success in the respective populations of heavily pre-treated patients with poor prognosis, and included long-lasting disease stabilisation (Cohorts 1–3), and prolonged survival (Cohort 4).

Secondary activity endpoints included: ORR, duration of response (DoR) defined as the time elapsed from the first response to PD; progression-free survival (PFS) according to irRECIST; and OS. Safety was based on the assessment of AEs, clinical laboratory test results, vital signs, and physical examination. AEs and laboratory values were graded according to the NCI-CTCAE v. 4.03. The trial included an exploratory analysis to correlate the expression of PD-L1 CPS and activity endpoints.

## Statistical considerations

The sample size was calculated using a one-sample superiority test, function One Sample Proportion NIS of the Trial Size package of R software (version 3.6.3 [2020-02-29] "Holding the Windsock". The R Foundation for Statistical Computing, Vienna, Austria). For Cohorts 1 to 3, according to previous reports, it was assumed that the 9-m CBR was 30% (null hypothesis) and a potential 20% increase was estimated with a superiority margin of 10% (alternative hypothesis). For cohort 4, according to previous reports[12–14], it was assumed that the 9-m OS rate was 13% (null hypothesis) and a 10% increase with a superiority margin of 5% (alternative hypothesis) was estimated. With a unilateral alpha level of 5 and 80% power and a 10% loss to follow-up rate, the required sample size was: 31 in Cohorts 1 to 3 and 33 in Cohort 4 (126 patients in total).

Activity analysis was based on the full analysis set (FAS) that included all enrolled patients. Safety was assessed for all patients who received at least one dose of study treatment. Data after the first progression for the three retreated patients was excluded from the current analysis. Continuous variables were summarised using descriptive statistics. Frequency counts and the percentage of subjects were provided for categorical data. Response rates were estimated using 95% confidence intervals (CI) or full range intervals. The survival or time-to-event endpoints were estimated using the Kaplan–Meier method. Cohen's kappa test was used to compare the distribution of best objective responses and measures the agreement between RECIST and iRECIST assessment. Patients without documented progression or death at the time of the analysis were censored at the last date of tumour evaluation for PFS assessment. Patients without documented death at the time of the analysis were censored at the last date of follow-up for OS assessment. All statistical tests were considered two-tailed, and results with $p < 0.05$ were considered significant. Data were collected through the MFAR eCRF system. All statistical analyses were performed with R and SPSS (IBM SPSS Statistics Version 26, Armonk, NY). Figures and tables were generated using RStudio (Version 1.2.5033 2009-2019 RStudio, Inc., Boston, MA, US).

## Reporting summary

Further information on research design is available in the Nature Portfolio Reporting Summary linked to this article.

## Data availability

The study protocol is available as Supplementary Note in the Supplementary Information file. The raw data are protected and are not available due to data privacy laws. The data that support the findings of this study are available from the corresponding author upon request (equivalent purposes to those for which the patients grant their consent to use the data: i.e. for research in neuroendocrine neoplasms). Data will be provided anonymously, with no identifiable data. The remaining data are available within the Article, Supplementary Information or Source Data file. Source data are provided with this paper.

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

## Acknowledgements

This work was supported by the Grupo Español de Tumores Neu-roendocrinos y Endocrinos (GETNE). AstraZeneca provided durvalumab and tremelimumab and awarded a grant to GETNE to pay the costs of the study. The funder did not have a role in designing or conducting the study. The authors thank all patients and families, investigators and study staff involved in the DUNE trial; the MFAR Clinical Research team for regulatory, monitoring, and quality assurance activities; Pau Doñate PhD for manuscript and language editing; and Jordi Curto M.Sc. and Arturo Alvarez PhD for statistical support.

## Author contributions

J.C. contributed to study design and coordination. These authors con-tributed equally to accrual, study conduct and manuscript editing to the current manuscript: J.C., J.H., A.T., C.L., R.G-C., M. Be., A. Cus., A. G.-A., A. Cub., V.A., A.C.-B., T.A.-G., G.C., P.J.-F., M. Bl., A.V., A.L.C., I.S., A.S., M.L., S.L., P.N., and J.L.M.

## Competing interests

J.C. has participated as a scientific consultant for Amgen, Bayer, Eisai, AAA, Ipsen, Pfizer, Merck, Sanofi, Novartis, Lilly, ITM, Hutchinson Pharma and Exelixis; has received grants for research purposes from Eisai, AstraZeneca, AAA, Ipsen, Pfizer and Novartis; and is currently chair of the GETNE group, and advisory member of the European Neuroendocrine Tumour Society (ENETS) society. J.H. has participated as a speaker on behalf of Eisai, Ipsen, Novartis, AAA and Angelini. A.T. has participated as a speaker, advisor and received travel grants from Novartis, Ipsen, AAA, Pfizer and AstraZeneca. R.G-C. has participated as a scientific consultant for AAA, Advanz Pharma, Bayer, BMS, HMP, Ipsen, Merck, Midatech Pharma, MSD, Novartis, PharmaMar, Pfizer, Pierre Fabre, Roche, Sanofi and Servier; has received grants for research purposes from Pfizer, BMS and MSD; received institutional financial support from ARMO Biosciences, Astrazeneca, Pfizer, Novartis, Ipsen, Roche, Pharmacyclics, Boston Bio-medicals, Merck, MSD, Amgen, Sanofi, Bayer, Bristol-Myers-Squibb, Boerhringer, Sysmex, Gilead Sciences, Servier, Adacap, VCN, Lilly and Pharmamar; and is currently a member of the Executive Committee of GETNE and ENETS, and member of the EORTC, ASCO, ESMO, SEOM, TTD, GEMCAD working groups. T.A.-G. has participated as a speaker on behalf of Ipsen, Pfizer, Adacap, BMS and Eisai; has participated as a scientific consultant for Ipsen, Pfizer, BMS, Eisai, Roche, Sanofi, Bayer, Astellas, Janssen-Cilag, Adacap and Lilly; has received grants for research pur-poses from Ipsen, Pfizer and Roche; has participated as scientific advisor at the Spanish Society of Medical Oncology (SEOM). G.C. has participated as a scientific consultant or speaker for BMS, MSD, Ipsen, Roche, Eisai, Sanofi, Janssen, Eusa Pharma and Pierre Fabre. I.S. has participated as a scientific consultant for Ipsen, Pfizer, AAA, Syrtex, Avanz Pharma, Phar-mamar and Amgen; has participated as a speaker on behalf of Ipsen, Pfizer, Pierre Fabre, Pharmamar, Bayer and AAA. C.L. has participated as a scientific consultant for Ipsen, Roche, Eisai, Novartis, Pfizer, AstraZeneca, Pierre Fabre, Sanofi, Bayer, Servier and AAA; has received grants for research purposes from Ipsen, Roche, Pfizer, AstraZeneca, Pierre Fabre and Novartis. M.B. has participated as a speaker on behalf of Pfizer, Ipsen and Novartis. P.N. has participated as a scientific consultant for Bayer; and as a scientific consultant and speaker on behalf of MSD and Novartis. M.L. has participated as a scientific consultant for IPSEN, INCYTE, AAA; and as a speaker on behalf of AMGEN, SERVIER, IPSEN, PIERRE FABRE, AND AAA. V.A. has participated as a speaker on behalf of Ipsen, Amgen, Merck, Servier, AAA and Pierre Fabre; has participated as a scientific consultant for Ipsen, Roche, Sanofi and Merck. A.G.-A. has participated as a speaker on behalf of Angelini Pharma Spain; and received travel accommodation expenses from Pfizer, Ipsen and Eisai Europe. A.V. is employed by ICON plc. The remaining authors declare no competing interests.

## Additional information

J. Capdevila [1,2] ✉, J. Hernando[1], A. Teule [3], C. Lopez[4], R. Garcia-Carbonero[5], M. Benavent[6], A. Custodio[7], A. Garcia-Alvarez [1], A. Cubillo[8], V. Alonso[9], A. Carmona-Bayonas[10], T. Alonso-Gordoa[11], G. Crespo [12], P. Jimenez-Fonseca[13], M. Blanco[14], A. Viudez[15], A. La Casta[16], I. Sevilla[17], A. Segura[18], M. Llanos[19], S. Landolfi [20], P. Nuciforo [21] & J. L. Manzano[22]

[1]Medical Oncology Department, Vall Hebron University Hospital, Vall Hebron Institute of Oncology (VHIO), Barcelona, Spain. [2]Medical Oncology Department, IOB-Quiron-Teknon, Barcelona, Spain. [3]Medical Oncology Department, Institut Català d'Oncologia (ICO) - IDIBELL L'Hospitalet del Llobregat, L'Hospitalet de Llobregat, Spain. [4]Medical Oncology Department, Hospital Universitario Marqués de Valdecilla, IDIVAL, Santander, Spain. [5]Medical Oncology Department, Hospital Universitario 12 de Octubre, Imas12, UCM, CNIO, Madrid, Spain. [6]Medical Oncology Department, University Hospital Virgen del Rocío, Instituto de Biomedicina de Sevilla (IBIS), Seville, Spain. [7]Medical Oncology Department, Hospital Universitario La Paz, Madrid, Spain. [8]Medical Oncology Department, Hospital Universitario HM Sanchinarro, Madrid, Spain. [9]Medical Oncology Department, Hospital Universitario Miguel Servet, Instituto de Investigación Sanitaria de Aragón (IISA), Zaragoza, Spain. [10]Hematology and Medical Oncology Department, Hospital Universitario Morales Meseguer, UMU, IMIB, Murcia, Spain. [11]Medical Oncology, Hospital Universitario Ramón y Cajal, Madrid, Spain. [12]Medical Oncology Department, Complejo Asistencial Universitario de Burgos, Burgos, Spain. [13]Medical Oncology Department, Hospital Universitario Central de Asturias, ISPA, Oviedo, Spain. [14]Medical Oncology Department, Hospital Universitario Gregorio Marañón, Madrid, Spain. [15]Medical Oncology Department, Hospital Universitario de Navarra, Pamplona, Spain. [16]Medical Oncology Department, Hospital Universitario Donostia, San Sebastián, Spain. [17]Medical Oncology Department, Investigación Clínica y Traslacional en Cáncer/Instituto de Investigaciones Biomédicas de Málaga (IBIMA)/Hospitales Universitarios Regional y Virgen de la Victoria de Málaga, Málaga, Spain. [18]Medical Oncology Department, Hospital Universitario y Politécnico La Fe, Valencia, Spain. [19]Medical Oncology Department, Hospital Universitario de Canarias, San Cristobal de la Laguna, Spain. [20]Pathology Department, Vall Hebron University Hospital, CIBERONC, Barcelona, Spain. [21]Molecular Oncology Group. Vall Hebron Institute of Oncology (VHIO), Barcelona, Spain. [22]Medical Oncology Department, Institut Català d'Oncologia (ICO) - Badalona, Hospital Germans Trias i Pujol, Badalona, Spain. ✉e-mail: jcapdevila@vhio.net

