## [Peer Review File · Nature Communications]

Durvalumab plus tremelimumab for the treatment of advanced neuroendocrine neoplasms of gastroenteropancreatic and lung originREVIEWER COMMENTS

Reviewer #1 (Remarks to the Author): with expertise in cancer immunotherapy, clinical

Capdevilla et al report on the so far largest cohort of patients with advanced NENs treated with immunotherapy using combined checkpoint blockade.

The authors should be congratulated for having conducted a well-designed multi-centre clinical trial across one country (Spain) in a group of patients with rare malignancies.

As the authors correctly concluded the clinical activity of the Durvalumab/Tremelimumab regimen in DUNE was overall modest although a signal of activity has been observed in patients with high grade neuroendocrine neoplasms that may warrant further investigation.

Comments to be addressed:

Introduction

101-102: References 15/16 are not appropriate as these refer to a small phase II trial in patients with ocular melanoma and a phase III lung cancer trial which cannot be seen as practice changing; suitable references include any of the publications reporting on the outcomes of combined checkpoint blockade in patients with cutaneous melanoma (CheckMate067) and/or renal cell carcinoma (CheckMate214).

109-111: Given that there are currently very limited data sets on the treatment of patients with advanced neuroendocrine cancers using combined checkpoint blockade, an Australian clinical trial using ipilimumab and nivolumab should be referenced (Klein O et al, Clin Cancer Res 2020).

Results

118-119: The exact anatomical location of NENs enrolled into cohort 2 and 4 should be provided in the table, in addition for cohort 4 the exact histological diagnosis (large cell, small cell etc) should be included

122-146: Reporting on the response rate and survival endpoints for the whole study cohort (FAS) is not meaningful given the heterogeneity in the clinical course/survival between low grade and high NEN and should be deleted.

148-156: It would be helpful to classify and group adverse events as immune related adverse events (irAEs) and state the overall percentage of severe high grade immune related toxicity, the current text implies that around ~10% of patients developed high grade immune related toxicity requiring treatment discontinuation (Line 150: unacceptable toxicity?)

156: It is unclear to the reader what "diarrhoea with encephalitis infection" represents? Immune mediated entero-colitis and encephalitis?

Discussion

165-168: The authors refer to trial data suggesting a higher response rate with anti-PD-1 based checkpoint blockade in patients with lung NENs.

The Spatalizumab trial (Yao JC et al.) reported a response rate of 17% in the thoracic cohort (also included thymic NETs so response higher in lung NETs), the authors referenced this trial (Ref 19 as a meeting abstract), the trial results have in the meantime been published as a full manuscript (Endocrine related cancers, 2021) and it should be referenced accordingly; similarly to the Spatalizumab trial (all lung NET responders atypical carcinoid), the Australian trial (Klein O et al, 2020) reported a response rate of 30% in patients with atypical carcinoid.

DUNE enrolled six patients with atypical carcinoid, it would be interesting to know if any of the three

responders of cohort 1 had an atypical carcinoid?

168-174: The authors correctly caution to compare outcomes across different trials (Dune and DART/CA209-538); a discussion point that could be elaborated on and may account for differences in the trial outcomes seen is that dual checkpoint blockade using Durvalumab/Tremelimumab does not equal Nivolumab/Ipilimumab therapy, the former regimen includes an anti-PD-L1 antibody, the latter an anti-PD-1 antibody and given additional receptor-ligand interactions, immunological and clinical outcomes may differ between these treatment regimens. In addition, there are likely differences in efficacy between the CTLA-4 antibodies Tremelimumab and Ipilimumab based on dosing/dosing interval and Fc-Isotype of the antibodies.

184-189: The exact histological diagnoses (according to WHO classification) of the responders, patients with prolonged disease control would be of interest; did any of the responding patients had a microsatellite unstable tumour?

202-203: The statement is incorrect, the listed reference refers to a chemo-immunotherapy trial in SCLC, comparisons in regards to safety should be made to Durvalumab/Tremelimumab combination trials in other malignancies using the same dosing schedule.

The second sentence reads that premature trial discontinuation has been due to toxicity in the majority of patients, however only 10% of patients discontinued treatment due to side effects according to the result section? If high grade immune related toxicity has only been seen in around 10% of patients, this would be very low for combined checkpoint blockade.

As mentioned above, it would be valuable to state the overall frequency of immune related toxicity and grade 3/4 immune related toxicity as this will provide the reader with a better appreciation of the immune-activation potential of the Durvalumab/Tremelimumab regimen; a low rate of high grade immune related toxicity may imply that the regimen has insufficient potential to successfully re-invigorate an anti-tumour immune response explaining limited clinical activity. Of note, the 1mg Ipi/3mg Nivo regimen leads to high grade irAEs in around a third of treated patients.

Reviewer #2 (Remarks to the Author): with expertise in biostatistics, clinical trial study design

Please see detailed comments and concerns below.

1. P5L126: Provide the full name of FAS for the first time use. FAS was defined on page 10.

2. P5L130-131: Fig.S1 was referred after the results of the duration of response (DoR) and the duration of SD. However, DoR or duration of SD for all patients is hard to get from Fig. S1. Fig. S1 is a Spider plot showing changes in tumor size from baseline for each patient. DoR or duration of SD is hard to get because the time of response or time of progression could not find in the plots. Kaplan-Meier curve or swimmer's plot could better present the results for DoR and duration of SD. Moreover, it would be better also to report 95% CI of median duration if not all the patients are progressed yet.

3. P5L141: It would be better to add 95%CI for the median OS in lung NETs (Cohort 1) even though the median OSU is not achieved. Also, add the 95%CI for the median OS for cohort 1 in Figure 3.

4. P5L143-144: It would be better to inform the audience and indicate the pre-established futility threshold here.

5. P10L299-300: Distinguish between the multivariate analyses and multivariable analysis. Multivariate regression models refer to statistical models with 2 or more dependent or outcome variables, and multivariable regression models refer to statistical models with multiple independent variables. So the multivariable regression model is the more appropriate name in this manuscript. Moreover, if multiple independent variables were included in the model, it should be indicated in the results. If no results were from multivariable regression model in the manuscript, this sentence should

be deleted.

6. P10L301: Cox regression analysis was not used in this manuscript based on the results section. If so, it should be removed from the statistical considerations section.

7. P10: The statistical method used for testing the agreement (Cohen's kappa test) of RECIST 1.1 and irRECIST 1.1 should be stated in the statistical considerations section.

Reviewer #3 (Remarks to the Author): with expertise in neuroendocrine Tumors, clinical

This is a non-randomized phase II clinical trial assessing the activity and safety of the combination of the anti-PD-L1 durvalumab and the anti CTLA-4 tremelimumab in advanced neuroendocrine neoplasms (NENs). The patients' population (N=123) is stratified in four cohorts depending on the histology, grade and primary tumor origin (cohort 1 lung carcinoids, cohort 2 G1 and G2 gastrointestinal NETs, cohort 3 G1 and G2 pancreatic NET and, finally, cohort 4 G3 gastroenteropancreatic (GEP) NENs). The primary objectives are different according to the cohorts: for cohorts 1,2 and 3 (NET cohorts) was the 9-month disease control rate (DCR), whereas for cohort 4 was the 9-months overall survival (OS) rate.

The present study is interesting in terms of rationale, aims, and provided discussion. The methods, design and statistical plan are appropriate, properly conducted and reported.

Abstract accurately reports a summary of the research carried out. The conclusions drawn are supported by the data.

Overall, the manuscript is clear and well written. The topic is very refined. The results suggest a mild efficacy of this immune-combination. Notably, the authors evidenced a modest survival benefit in cohort 4. This aspect is quite relevant and, of course, deserves further dedicated studies.

I have few minor revisions. Should the authors achieve to answer them, I would strongly recommend publication.

1) Please, define "FAS" ("full analysis set") the first time that it is mentioned in the text (page 5 line 5).

2) I encourage the authors to carefully check all across the manuscript and correct little typos as for example:

- page 5 line 10 "(range: 0.9-30.7)(Fig.S1)" instead of "(range: 0.9-30.7) (Fig.S1)"

- page 6 line 1 "(23.6%)(Table2)" instead of "(23.6%) (Table 2)"

- page 6 line 1 and 2 uniform the font

- page 23 uniform the font of the Table 2 heading

3) Please, spell out the acronym "ICB" in the conclusion section, page 7 line 30.

4) In the discussion it is mentioned that "Long term survivors had mostly poorly differentiated NECs (70%) by central review; however no significant correlation was found between baseline molecular or clinical biomarkers and treatment in this subgroup (Table S2)." Among the supplementary material (Table S3) is reported the N and percentage of NET G3 vs NEC in relation to OS. It may be beneficial to add a more specific comment about this issue in the results as well as in the discussion section (due to the biological implications and difference between the two entities).

5) Finally, I would recommend the authors to add in the discussion a brief comment (and the corresponding reference) about the NICE NEC phase II study (a trial assessing the activity of the combination of nivolumab plus platinum-doublet chemotherapy in untreated advanced GEP or unknown origin G3 NENs).

Reviewer #1 (Remarks to the Author) with expertise in immunotherapy

Capdevilla et al report on the so far largest cohort of patients with advanced NENs treated with immunotherapy using combined checkpoint blockade.

The authors should be congratulated for having conducted a well-designed multi-centre clinical trial across one country (Spain) in a group of patients with rare malignancies.

As the authors correctly concluded the clinical activity of the Durvalumab/Tremelimumab regimen in DUNE was overall modest although a signal of activity has been observed in patients with high grade neuroendocrine neoplasms that may warrant further investigation.

Author response: We acknowledge your time for reviewing our research and thank you for all your comments that substantially improved the final manuscript.

Comments to be addressed:

Introduction

101-102: References 15/16 are not appropriate as these refer to a small phase II trial in patients with ocular melanoma and a phase III lung cancer trial which cannot be seen as practice changing; suitable references include any of the publications reporting on the outcomes of combined checkpoint blockade in patients with cutaneous melanoma (CheckMate067) and/or renal cell carcinoma (CheckMate214).

Author response: Following your advice, we have changed these references to the proposed phase III clinical trials. Thank you.

109-111: Given that there are currently very limited data sets on the treatment of patients with advanced neuroendocrine cancers using combined checkpoint blockade, an Australian clinical trial using ipilimumab and nivolumab should be referenced (Klein O et al, Clin Cancer Res 2020).

Author response: Following your advice, we have included the proposed reference.

Results

118-119: The exact anatomical location of NENs enrolled into cohort 2 and 4 should be provided in the table, in addition for cohort 4 the exact histological diagnosis (large cell, small cell etc) should be included

Author response: Following your advice, this information has been included in table 1 with patient characteristics. We had no patients with small cell type in the lung NETs cohort. Exact histological diagnosis for cohort 4 reviewed by central pathologist has been included within Table 1. Thank you so much for your suggestions.

122-146: Reporting on the response rate and survival endpoints for the whole study cohort (FAS) is not meaningful given the heterogeneity in the clinical course/survival between low grade and high NEN and should be deleted.

Author response: This data is commonly reported in smaller trials with heterogeneous populations of NENs. We agreed that meaningful results are those within each cohort and that is why the trial was designed and statistically powered by cohorts and results in figures are also by cohort. Following your advice, we removed the FAS endpoints.

148-156: It would be helpful to classify and group adverse events as immune related adverse events (irAEs) and state the overall percentage of severe high grade immune related toxicity, the current text implies that around ~10% of patients developed high grade immune related toxicity requiring treatment discontinuation (Line 150: unacceptable toxicity?)

Author response: Following your advice, we give more specific information regarding high grade irAEs:

Grade ≥ 3 immune-related AEs occurred in 12.2% of patients and included as the more common events hepatitis (1.6%), myositis (1.6%), and anaemia (1.6%).

All treatment discontinuations due to treatment related AEs do not imply an irAE. Please find enclosed a list of discontinuations reasons due to unacceptable toxicity:

Patient ID	Reason treatment discontinuation
04-003	Transaminitis G3
07-003	Diarrhoea G3
03-003	Anemia and fatigue G3
04-005	Colitis G3
01-015	Autoimmune hepatitis G4
13-001	Myocarditis G3
01-019	Creatinine increased G2
03-006	Arthralgia, Flu-like symptoms, myalgia, and fatigue G1
05-003	PRES, edema and mucositis G2
16-003	Rash G1
02-007	Hepatic failure G5
17-008	Miastenia gravis G5

156: It is unclear to the reader what “diarrhoea with encephalitis infection” represents? Immune mediated entero-colitis and encephalitis?

Author response: Following your advice, we tried to modify the description of the event to improve the intelligibility. The patient had a complex scenario with several symptoms. The event was classified by the principal investigator as diarrhoea with encephalitis. The more detailed description included: *The patient was hospitalized with a diarrhea grade 3 that worsened, developing acute renal failure, difficulties to answer and generalized clonic tonic crisis. Finally died during the mooring. suspected causes of death were diarrhea, renal failure and a possible encephalitis that could not be confirmed. Relationships with study drugs could not be discarded.*

Deaths caused by toxicity were associated with hepatic failure, myasthenia gravis, and diarrhoea that worsened and coursed with a potential encephalitis.

Discussion

165-168: The authors refer to trial data suggesting a higher response rate with anti-PD-1 based checkpoint blockade in patients with lung NENs.

The Spartalizumab trial (Yao JC et al.) reported a response rate of 17% in the thoracic cohort (also included thymic NETs so response higher in lung NETs), the authors referenced this trial (Ref 19 as a meeting abstract), the trial results have in the meantime been published as a full manuscript (Endocrine related cancers, 2021) and it should be referenced accordingly; similarly to the Spartalizumab trial (all lung NET responders atypical carcinoid), the Australian trial (Klein O et al,2020) reported a response rate of 30% in patients with atypical carcinoid.

DUNE enrolled six patients with atypical carcinoid, it would be interesting to know if any of the three responders of cohort 1 had an atypical carcinoid?

Author response: Following your advice, we have substituted the meeting abstract by the full results manuscript. We also included reference to the Australian trial within this section to reinforce the positivity of immunotherapy in lung NETs. However, we observed a limited activity and efficacy in lung NETs. We only reported 3 responses in cohort 1.

Following your suggestions we analyzed the ORR stratified and there was an enrichment of response in the atypical group of patients, despite a low number of events. Two (30%) responses in patients with atypical carcinoids and one (4,7%) in patients with typical carcinoids. This relevant information has been included in the manuscript:

Results section: In lung NETs, PR was reported in 2 (33.3%) patients with atypical carcinoids and 1 (4.7%) with typical carcinoid.

Discussion: Despite the low number of responses, we did observe a higher response rate in patients with atypical carcinoids in agreement with previous trials.

168-174: The authors correctly caution to compare outcomes across different trials (Dune and DART/CA209-538); a discussion point that could be elaborated on and may account for differences in the trial outcomes seen is that dual checkpoint blockade using Durvalumab/Tremelimumab does not equal Nivolumab/Ipilimumab therapy, the former regimen includes an anti-PD-L1 antibody, the latter an anti-PD-1 antibody and given additional receptor-ligand interactions, immunological and clinical outcomes may differ between these treatment regimens. In addition, there are likely differences in efficacy between the CTLA-4 antibodies Tremelimumab and Ipilimumab based on dosing/dosing interval and Fc-Isotype of the antibodies.

Author response: We agree on this rationale and following your explanation we have included a sentence in the discussion to consider that molecular targets may also have an impact on efficacy and explain differences across trials.

Moreover, durvalumab targets PD-L1 while nivolumab is an anti-PD-1 antibody and given additional receptor-ligand interactions, immunological and clinical outcomes may differ between these treatment regimens.

184-189: The exact histological diagnoses (according to WHO classification) of the responders, patients with prolonged disease control would be of interest; did any of the responding patients had a microsatellite unstable tumour?

Author response: This is an interesting point. We have analyzed upon this comment the MSI status from the 8 patients who respond to treatment. All analyzable patients had microsatellite stable tumors at baseline. Please find the results enclosed in the table below. Due to small numbers, correlative analysis with efficacy outcomes is not considered adequate.

Patient ID	Cohort	Microsatellite status
01-019	C1	MSS
05-004	C1	MSS
08-004	C1	MSS
03-005	C3	MSS
04-008	C3	MSS
01-004	C4	MSS
01-008	C4	Not sample available
17-004	C4	MSS

These results have been commented on results section:

Microsatellite instability was assessed in 7 out of 8 patients who had a response, being all microsatellite stable.

We did observe some response trends with tumor grade. Patients in cohort 4 (grade 3 GEP-NENs) had higher ORR than in cohort 2 or 3 (grade 1-2 GEP-NENs), however these differences are not significant due to the small number of responders and lack of statistical power. Differentiation also seems to correlate with activity, with higher ORR in NECs (cohort 4). In fact, one of our main conclusions is that durvalumab plus tremelimumab may have a use on high-grade GEP-NENs. Please find enclosed for your information a relation of patients with tumor response and their tumor grade:

Patient ID	Cohort	Tumor grade	Response
01-019	C1	G2	PR
05-004	C1	G2	PR
08-004	C1	G1	PR
03-005	C3	G1	PR
04-008	C3	G1	PR
01-004	C4	G3	PR
01-008	C4	G3	PR

17-004	C4	G3	CR
--------	----	----	----

202-203: The statement is incorrect, the listed reference refers to a chemo-immunotherapy trial in SCLC, comparisons in regards to safety should be made to Durvalumab/Tremelimumab combination trials in other malignancies using the same dosing schedule.

Author response: Following your advice, we have removed this reference from the manuscript and referenced two big phase III trial using the same combination.

The second sentence reads that premature trial discontinuation has been due to toxicity in the majority of patients, however only 10% of patients discontinued treatment due to side effects according to the result section? If high grade immune related toxicity has only been seen in around 10% of patients, this would be very low for combined checkpoint blockade.

Author response: Patients may discontinue treatment prematurely due to other reasons rather than toxicity (i.e. PD, patient withdrawal, non related AEs). We agree that the focus should be put in toxicity and, following your suggestion, we have modified this sentence:

Premature treatment discontinuation due to toxicity was required in about 10% of patients, which unlikely impacted activity. The relatively short treatment exposure may explain the low incidence of severe immune-related adverse events compared with other trials with combined checkpoint blockade.

There is a difference between toxicities that lead to treatment discontinuation, high grade toxicities (G3-5 events, which does not always end up with treatment discontinuation), and toxicities immune mediated (irAEs, which only include events secondary to study drugs and associated with immune system activity. Also, irAE does not necessarily end up with treatment discontinuation). We had 9.8% of patients who discontinued due to toxicities, 29.3% who experienced Grade 3-5 toxicities and 12.2% who experienced Grade 3-5 irAEs. We have modified the results section for further clarity on the percentage of patients who experience G3-5 toxicities:

Grade ≥ 3 toxicities had low frequency (29.3% of patients), and the most common were diarrhoea (6.5%), transaminitis (4.9%), fatigue (3.3%), and vomiting (2.4%) (Table 2). Grade ≥ 3 immune-related AEs occurred in 12.2% of patients and included as the more common events hepatitis (1.6%), myositis (1.6%), and anaemia (1.6%).

As mentioned above, it would be valuable to state the overall frequency of immune related toxicity and grade 3/4 immune related toxicity as this will provide the reader with a better appreciation of the immune-activation potential of the Durvalumab/Tremelimumab regimen; a low rate of high grade immune related toxicity may imply that the regimen has insufficient potential to successfully re-invigorate an anti-tumour immune response explaining limited clinical activity. Of note, the 1mg Ipi/3mgNivo regimen leads to high grade irAEs in around a third of treated patients.

Author response: following your advice, we have included this data on the results section. The frequency of treatment-related adverse events was consistent with previous experience and also in line with other immune checkpoint inhibitors. We hypothesize that lack of overall efficacy is most probably related with the nature of these neoplasms than with lack of efficacy in terms of immune activation. We do see some groups of patients who could potentially benefit. In agreement with previous trials and thanks to your suggestions, patients with atypical lung carcinoids may be an interesting group to explore. Also high grade NENs, in which combination with platinum-based chemotherapy

Reviewer #2 (Remarks to the Author) with expertise in biostatistics, clinical trial study design

Please see detailed comments and concerns below.

1. P5L126: Provide the full name of FAS for the first time use. FAS was defined on page 10.

Author response: Following your advice, FAS has been defined for the first time use. As some results for FAS have been removed to satisfy reviewers requests the first appearance is in the methods section.

2. P5L130-131: Fig.S1 was referred after the results of the duration of response (DoR) and the duration of SD. However, DoR or duration of SD for all patients is hard to get from Fig. S1. Fig. S1 is a Spider plot showing changes in tumor size from baseline for each patient. DoR or duration of SD is hard to get because the time of response or time of progression could not find in the plots. Kaplan-Meier curve or swimmer's plot could better present the results for DoR and duration of SD. Moreover, it would be better also to report 95% CI of median duration if not all the patients are progressed yet.

Author response: In figure S1 time is represented on X axis. It could be seen the timepoint in which a patient reaches SD or response and the time until PD. Following your advice, we have modified the figure to better represent the DoR and Duration of SD, adding a Swimmer's plot.

Following your advice, the 95% CI has been added in the results section.

3. P5L141: It would be better to add 95%CI for the median OS in lung NETs (Cohort 1) even though the median OSU is not achieved. Also, add the 95%CI for the median OS for cohort 1 in Figure 3.

Author response: It is not possible to calculate the confidence interval of a median which could not be estimated (not reached). However, following your advice, the full range has been added in the results section.

4. P5L143-144: It would be better to inform the audience and indicate the pre-established futility threshold here.

Author response: Following your advice, the futility threshold, which was already specified in the methods section, is added in this section:

For high-grade GEP-NENs (Cohort 4), the 9-m OS rate, which was the primary endpoint, was 36.1% (95% CI: 19.6-52.6), surpassing the pre-established futility threshold (23%; H0= 13%).

5. P10L299-300: Distinguish between the multivariate analyses and multivariable analysis. Multivariate regression models refer to statistical models with 2 or more dependent or outcome variables, and multivariable regression models refer to statistical models with multiple independent variables. So the multivariable regression model is the more appropriate name in this manuscript. Moreover, if multiple independent variables were included in the model, it should be indicated in the

results. If no results were from multivariable regression model in the manuscript, this sentence should be deleted.

Author response: This section was used in a broader analysis including some multivariate models. However due to small sample size within each cohort the analysis was excluded from the final manuscript. Following your advice, the method has been removed.

6. P10L301: Cox regression analysis was not used in this manuscript based on the results section. If so, it should be removed from the statistical considerations section.

Author response: Following your observation, this information has been removed from the statistical considerations.

7. P10: The statistical method used for testing the agreement (Cohen's kappa test) of RECIST 1.1 and irRECIST 1.1 should be stated in the statistical considerations section.

Author response: Following your observation, this information has been added to the statistical considerations:

The survival or time-to-event endpoints were estimated using the Kaplan–Meier method. Cohen's kappa test was used to compare the distribution of best objective responses and measures the agreement between RECIST and irRECIST assessment.

Reviewer #3 (Remarks to the Author): with expertise in neuroendocrine Tumors, clinical

This is a non-randomized phase II clinical trial assessing the activity and safety of the combination of the anti-PD-L1 durvalumab and the anti CTLA-4 tremelimumab in advanced neuroendocrine neoplasms (NENs). The patients' population (N=123) is stratified in four cohorts depending on the histology, grade and primary tumor origin (cohort 1 lung carcinoids, cohort 2 G1 and G2 gastrointestinal NETs, cohort 3 G1 and G2 pancreatic NET and, finally, cohort 4 G3 gastroenteropancreatic (GEP) NENs). The primary objectives are different according to the cohorts: for cohorts 1,2 and 3 (NET cohorts) was the 9-month disease control rate (DCR), whereas for cohort 4 was the 9-months overall survival (OS) rate.

The present study is interesting in terms of rationale, aims, and provided discussion. The methods, design and statistical plan are appropriate, properly conducted and reported.

Abstract accurately reports a summary of the research carried out. The conclusions drawn are supported by the data.

Overall, the manuscript is clear and well written. The topic is very refined. The results suggest a mild efficacy of this immune-combination. Notably, the authors evidenced a modest survival benefit in cohort 4. This aspect is quite relevant and, of course, deserves further dedicated studies.

Author response: The authors want to acknowledge your interest in our research. Thank you for all your comments that substantially improved the final manuscript.

I have few minor revisions. Should the authors achieve to answer them, I would strongly recommend publication.

1) Please, define "FAS" ("full analysis set") the first time that it is mentioned in the text (page 5 line 5).

Author response: Following your advice, FAS has been defined at the first time it is mentioned.

2) I encourage the authors to carefully check all across the manuscript and correct little typos as for example:

- page 5 line 10 “(range: 0.9-30.7)(Fig.S1)” instead of “(range: 0.9-30.7) (Fig.S1)”
- page 6 line 1 “(23.6%)(Table2)” instead of “(23.6%) (Table 2)”
- page 6 line 1 and 2 uniform the font
- page 23 uniform the font of the Table 2 heading

Author response: Following your advice, the manuscript has been checked thoroughly and all the abovementioned typos have been corrected.

3) Please, spell out the acronym “ICB” in the conclusion section, page 7 line 30.

Author response: ICB stands for Immune checkpoint blockade. This is equivalent to immune checkpoint inhibition (ICI), which is used throughout the manuscript. Please excuse the duplicity. Following your suggestion, we have modified the acronym to ICI for consistency.

4) In the discussion it is mentioned that “Long term survivors had mostly poorly differentiated NECs (70%) by central review; however no significant correlation was found between baseline molecular or clinical biomarkers and treatment in this subgroup (Table S2).” Among the supplementary material (Table S3) is reported the N and percentage of NET G3 vs NEC in relation to OS. It may be beneficial to add a more specific comment about this issue in the results as well as in the discussion section (due to the biological implications and difference between the two entities).

Author response: Long term survivors are enriched modestly on NEC (not statistically significant). This is a relevant finding which suggests that poor differentiation could correlate with prediction of ICI benefit. In fact, all responses in cohort 4 were reported in patients with NEC. Similar results are observed with lung NETs (atypical vs typical carcinoids). However the low number of patients will not allow to demonstrate this. Following your advice, we have mentioned this within the results and discussion sections:

In patients with grade 3 NENs, NECs had an ORR of 16.7% (3/18 patients) versus 0% in NETs.

A stratified analysis for OS within grade 3 GEP-NENs found no significant associations between survival status and baseline characteristics, including histological differentiation (NET vs NEC), or PD-L1 CPS status.

Our results showed no enrichment of activity regarding histological grade, with modest activity also in high-grade NENs (ORR 9.1%), with all responses occurring in patients with NECs.

5) Finally, I would recommend the authors to add in the discussion a brief comment (and the corresponding reference) about the NICE NEC phase II study (a trial assessing the activity of the combination of nivolumab plus platinum-doublet chemotherapy in untreated advanced GEP or unknown origin G3 NENs).

Author response: Following your advice, the abovementioned clinical trial has been referenced. We agree that high grade NENs could be those who benefit the most from immunotherapy according to our results. Additionally the combination with chemotherapy, which is known to induce generation of neoantigens and increase tumor immunogenicity, is a very interesting scenario to explore.

REVIEWERS' COMMENTS

Reviewer #1 (Remarks to the Author):

The authors have very well addressed reviewer 1+3 comments

Reviewer #2 (Remarks to the Author):

Thanks for addressing the comments.

RESPONSE TO REVIEWERS MARCH 8th 2023

Reviewer #1 (Remarks to the Author):

The authors have very well addressed reviewer 1+3 comments

Author response: We acknowledge the revision of our work and really appreciate your contributions to the final version.

Reviewer #2 (Remarks to the Author):

Thanks for addressing the comments.

Author response: We acknowledge the revision of our work and really appreciate your contributions to the final version.